# The Laccase Family Gene *CsLAC37* Participates in Resistance to *Colletotrichum gloeosporioides* Infection in Tea Plants

**DOI:** 10.3390/plants13060884

**Published:** 2024-03-19

**Authors:** Dangqiang Li, Hongxiu Zhang, Qianqian Zhou, Yongning Tao, Shuangshuang Wang, Pengke Wang, Aoni Wang, Chaoling Wei, Shengrui Liu

**Affiliations:** 1State Key Laboratory of Tea Plant Biology and Utilization, Anhui Agricultural University, Hefei 230036, China; 18356508647@163.com (D.L.); zhanghongxiu0513@163.com (H.Z.); zqq667221@163.com (Q.Z.); taoyongning3013@163.com (Y.T.); wangpengke@163.com (P.W.); 17375338632@163.com (A.W.); weichl@ahau.edu.cn (C.W.); 2Tea Research Institute, Shandong Academy of Agricultural Sciences, Jinan 250100, China; wangshuang0103@163.com

**Keywords:** *Camellia sinensis*, *CsLAC37*, anthracnose, fungal pathogens, functional validation

## Abstract

Fungal attacks have become a major obstacle in tea plantations. *Colletotrichum gloeosporioides* is one of the most devastating fungal pathogens in tea plantations that can severely affect tea yield and quality. However, the molecular mechanism of resistance genes involved in anthracnose is still largely unknown in tea plants. Here, we found that the laccase gene *CsLAC37* was involved in the response to fungal infection based on a transcriptome analysis. The full-length CDS of *CsLAC37* was cloned, and its protein sequence had the closest relationship with the *Arabidopsis* AtLAC15 protein compared to other AtLACs. Tissue-specific expression analysis showed that *CsLAC37* had higher expression levels in mature leaves and stems than in the other tissues. Subcellular localization showed that the CsLAC37 protein was predominantly localized in the cell membrane. The expression levels of *CsLAC37* were upregulated at different time points under cold, salt, SA, and ABA treatments. qRT-PCR confirmed that *CsLAC37* responded to both *Pestalotiopsis*-like species and *C. gloeosporioides* infections. Functional validation showed that the hydrogen peroxide (H_2_O_2_) content increased significantly, and POD activity decreased in leaves after antisense oligonucleotide (AsODN) treatment compared to the controls. The results demonstrated that *CsLAC37* may play an important role in resistance to anthracnose, and the findings provide a theoretical foundation for molecular breeding of tea varieties with resistance to fungal diseases.

## 1. Introduction

Tea plants (*Camellia sinensis*) originated in southwest China and are one of the most important woody economic crops worldwide. The tea industry belongs to a high-output and distinctive agriculture industry in China [1]. However, tea plants frequently suffer from various stresses, including drought, cold, insects, and diseases, seriously affecting their growth and development. Anthracnose is a widespread disease in tea plantations in China, which can severely affect the yield and quality of the tea [2]. *Colletotrichum gloeosporioides* is one of the dominant endophytic taxa of *C. sinensis*, which causes the fungal disease anthracnose [3]. The infection of *C. gloeosporioides* initially showed a biotrophic phase, followed by a necrotrophic-stage pathogen, leading to defoliation by affecting both young and old leaves [4].

Various methods, such as prevention and treatment, have been applied to control anthracnose disease in tea plantations. Chemical treatment is one of the most widely used methods for anthracnose disease control [5]. For instance, the Bordeaux mixture, chlorothalonil, carbendazim, and thiophanate methyl are effective chemicals that have been extensively used in tea plantations [6]. However, the excessive use of synthetic fungicides can lead to the emergence of pathogens resistant to fungicides and harmful effects on human health and the environment [7]. In comparison, breeding and cultivating disease-resistant tea varieties is an effective and environmentally friendly approach for controlling anthracnose disease. Nevertheless, there are few studies on tea plant disease resistance-related genes. Herein, to lay a solid foundation for understanding the resistance mechanisms, it is imperative to explore and characterize resistance-related genes in tea plants.

Laccase is the largest subfamily of multicopper oxidases that occurs widely in fungi, bacteria, insects, and plants [8,9]. Laccase in plants contains three cupredoxin domains that combine with four copper ions and have a broad spectrum of substrates [10]. Studies have shown that the laccase family genes play critical roles in plants, involving the polymerization of phenolic compounds, lignifying the cell wall structure, and defending against various stresses [11]. In *Arabidopsis thaliana*, *AtLAC4* and *AtLAC17* were involved in the constitutive lignification of stems; the lac4 and lac17 double mutants displayed hypolignified fibers and collapsed xylem vessel phenotypes, and the lac4, lac7, and lac11 triple mutants showed obvious growth defects, and the lignin content decreased significantly [12,13]. In *Brachypodium distachyon*, *BdLAC5* is required for the lignification of the *B. distachyon* culms; the mutant showed a 10% decrease in Klason lignin content, and the amount of ferulic acid units of ester in the cell walls increased by 40% [14]. In *Cleome hassleriana*, *ChLAC8* is critical for C-lignin polymerization and facilitates the polymerization of caffeyl alcohol for C-lignin biosynthesis [15]. In *Gossypium hirsutum*, *GhLAC15* enhances resistance to *Verticillium dahlia* infections by increasing the defense-induced lignification and lignin components in the cell walls [16]. In *Actinidia chinensis*, a genome-wide identification of the laccase gene family was performed, and several members showed significantly higher expression levels than the control after inoculation with *Pseudomonas syringae* pv. *actinidiae* [17]. Overall, these studies demonstrated that laccases had a critical role in plant development and stress responses by regulating lignin biosynthesis. Previously, we performed a comprehensive analysis of the laccase gene family in tea plants [8]. However, little is known about the biological functions of *CsLACs*, especially their roles in disease resistance.

Based on our previous studies, we found that the laccase gene *CsLAC37* was involved in the response to fungal infections in tea plants [8,18]. Therefore, we performed experiments to gain insights into its functions. We found that the expression level of *CsLAC37* increased under both *Pestalotiopsis-*like species and *C. gloeosporioides* infections. Moreover, *CsLAC37* was also involved in response to SA and ABA stresses. Antisense suppression (AsODN) in tea plants indicates that *CsLAC37* may play an important role in response to anthracnose disease. This study helps us understand the biological function of *CsLAC37* and provides a candidate gene resource for tea plant breeding.

## 2. Results

### 2.1. Identification of CsLAC37 and Protein Sequence Analysis

Based on our previous study, we found that *CsLAC37* from the laccase gene family was involved in responses to fungus infections [8]. Thereby, *CsLAC37* was selected as the candidate gene for its biological function in tea plants. The full-length cDNA sequence was cloned and sequenced, demonstrating that the full-length ORF of *CsLAC37* was 1701 bp, and the protein had a molecular weight of 68.1 kDa and isoelectric point (pI) value of 8.43. Phylogenetic analysis showed that its protein sequence had the highest similarity to AtLAC14 and AtLAC15 compared to the other AtLACs in *Arabidopsis* (Figure 1A). The conserved domains of the three proteins were further analyzed, demonstrating that all of them contained three conserved cupredoxin motifs (Cu-oxidase_1, Cuoxidase_2, and Cu-oxidase_3) (Figure 1B).

### 2.2. Tissue-Specific Expression Patterns and Subcellular Localization

Expression levels of *CsLAC37* in eight tissues (bud, the first leaf, the second leaf, the third leaf, stem, root, flower, and fruit) were analyzed (Figure 2A). Transcriptome data showed that *CsLAC37* had the highest expression levels in stems and fruits, followed by buds and leaves, while it had relatively low expression levels in roots and flowers (Figure 2B) [19]. Similarly, qRT-PCR verification showed that *CsLAC37* had the highest expression level in stems, followed by leaves, and had the lowest expression levels in roots and flowers (Figure 2C). The results showed that *CsLAC37* had tissue-specific expression patterns in tea plants.

To further understand the molecular function of the CsLAC37 protein, we transferred the CsLAC37-GFP recombinant plasmid into Agrobacterium cells to infect tobacco leaves, and the GFP empty plasmid was the control. A bright field showed the cell contour of the tobacco leaves, and mCherry was the positive control. The results showed that the CsLAC37 protein was mainly localized in the cell membrane (Figure 3).

### 2.3. Expression Analysis in Response to Fungal Infection

To understand the potential function of *CsLAC37* in response to fungal infections, we detected its expression levels under both grey blight and *C. gloeosporioides* infections. The expression levels of *CsLAC37* increased significantly at 7 d and 13 d under grey blight infection based on transcriptome data, while it decreased dramatically at 4 d. In comparison, a similar expression pattern of *CsLAC37* was observed from qRT-PCR validation (Figure 4A). Based on our previous transcriptome data, the expression level of *CsLAC37* can also be affected by *C. gloeosporioides* infections, demonstrating that the *CsLAC37* expression level increased significantly at 10 d after *C. gloeosporioides* infections. The qRT-PCR results also confirmed that the *CsLAC37* expression level increased significantly at 10 d, demonstrating a similar result compared to the transcriptome data (Figure 4B).

### 2.4. Expression Analysis in Response to Cold and Salt Stresses

Studies have shown that LAC genes respond to various abiotic stresses, such as cold and drought treatments. Thus, we detected the expression levels of *CsLAC37* in tea leaves at 12 h and 24 h under cold (0 °C) treatment. The *CsLAC37* expression level increased significantly at 24 h under cold treatment compared to the control (25 °C) (Figure 5A). Furthermore, we analyzed its expression level under salt stress based on transcriptome data [20], demonstrating that *CsLAC37* had a higher expression level under NaCl treatment than the control.

### 2.5. Expression Analysis in Response to SA and ABA Treatments

To elucidate whether *CsLAC37* was involved in response to hormones, we detected its expression levels under SA and ABA treatments at different time points. Under SA treatment, its expression level increased significantly at 9 h and 12 h compared to the control (Figure 6A), while no significant variation was observed at the other time points. Under ABA treatment, its expression level increased significantly at 9 h, while no significant variation was found at the other time points (Figure 6B). The results indicate that *CsLAC37* was involved in the response to both SA and ABA hormones.

### 2.6. Functional Validation of CsLAC37 in Resistance to Fungal Infection

To elucidate the molecular function of *CsLAC37* in resistance to fungal infections, antisense oligonucleotides (AsODNs) were implemented in shoots (one bud and three leaves). The efficiency of three probes was initially screened, demonstrating that all of them (AsODN-1, AsODN-2, and AsODN-3) suppressed the expression of *CsLAC37* effectively (Figure 7A). Subsequently, AsODN-1 was selected to screen optimal time points for AsODN at 24 h and 48 h. The results showed that the expression level of *CsLAC37* was suppressed significantly at 24 h compared to the controls (H_2_O and sODN), while no significant difference was observed at 48 h (Figure 7B).

After AsODN treatment for 24 h, the second leaf was infected by *C. gloeosporioides*. As shown in Figure 8A, lesions on the leaves were observed under normal conditions (upper left corner), fluorescent light (upper right corner), DAB staining (lower left corner), and NBT staining (lower right corner). However, no significant difference was observed in the lesion morphology and area on the leaves among the different treatments (H_2_O, sODN, and AsODN) (Figure 8A,B). Furthermore, three critical biochemical parameters including the H_2_O_2_ content, POD activity, and SOD activity were measured. Although the SOD activity varied insignificantly, the H_2_O_2_ content increased clearly, and the POD activity decreased significantly (Figure 8C–E). The results showed that the leaves were damaged more severely, and POD activity decreased after AsODN treatment of CsLAC37 compared to the controls, indicating that *CsLAC37* plays an important role in resistance to anthracnose disease.

## 3. Discussion

The plant cell wall is mainly composed of several pivotal components including cellulose, hemicelluloses, pectic polysaccharides, and lignin, which are essential in the interactions between plants and pathogens [16]. Lignin is the second-most abundant biopolymer, which has been reported to be involved in resistance to various stresses. Laccase is the largest subfamily of multicopper oxidases, which can catalyze the oxidative polymerization of lignin precursors [11]. Studies have confirmed that laccase genes play an important role in lignin biosynthesis in various plants. Although studies on genome-wide analysis of the laccase gene family have been performed in tea plants [8,9], little is known about their biological functions. In the current study, we identified and characterized a laccase gene *CsLAC37*, which was involved in the response to fungal infections. Additionally, we found that *CsLAC37* also participated in the response to drought and cold stresses [8]. These findings indicate that *CsLAC37* may play a critical role in resistance to stresses in tea plants.

The tissue expression pattern of *CsLAC37* can provide significant references for its biological functions. Laccases are key genes involved in lignin biosynthesis; thus, it is not surprising that *CsLAC37* was predominantly expressed in stems and mature leaves. Previous studies showed that different members from the laccase gene family have different tissue expression patterns in tea plants [8,9]. In other plants, it was common that many laccase members had distinct tissue expression patterns, such as *Arabidopsis* [21], *Pyrus bretschneideri* [22], and *Phyllostachys edulis* [23]. Moreover, subcellular localization showed that CsLAC37 was predominantly localized in the cell membrane. Our previous study showed that CsLAC3 was also localized in the cell membrane based on subcellular localization analysis using tobacco leaves [8]. During the early stage of Casparian strip development in *Arabidopsis*, *AtLAC3* was shown to prioritize expression in endoderm cells and penetrate the cell membrane to reach the lignin domain and cell wall boundary [24]. AtLAC4, which contributes to stem lignification, was localized in the lignified secondary cell wall [25]. In *Miscanthus*, it was shown that MsLAC1 was localized in the cell wall matrix [26]. Based on the results from our study and previous studies, we speculate that CsLAC37 may mainly have functions in the cell membrane.

Phylogenetic analysis showed that CsLAC37 had the highest sequence similarity with AtLAC14 and AtLAC15 in *Arabidopsis* than the other AtLACs. In *Arabidopsis*, the *lac15* (tt10) mutant seeds accumulated more epicatechin monomers and soluble proanthocyanidins, exhibiting a delay in the developmentally determined browning of the seed coat [27], and its mutant seeds had an almost 30% decrease in lignin content than in wild-type seeds [28]. The *GhLAC15* from *Gossypium hirsutum* was phylogenetically related to *AtLAC15*, and overexpression of *GhLAC15* enhanced cell wall lignification, which significantly improved the *Verticillium wilt* resistance of transgenic *Arabidopsis* [16]. Thus, *CsLAC37* may also play an important role in tea plant leaves and stems by regulating lignin biosynthesis.

SA functions as a pivotal signal in regulating disease resistance in plants. SA and its derivative methyl salicylate (MeSA) activate both local and systemic acquired resistance [29]. In rice, Phenylalanine ammonia-lyases (OsPALs) mediated resistance to brown planthoppers by regulating the biosynthesis and accumulation of SA and lignin, and the expression of PALs was positively regulated by *OsMYB30* [30]. In *Medicago sativa*, SA levels were inversely proportional to the lignin levels and growth in many transgenic alfalfa plants, where lignin biosynthesis was disturbed [31]. After sulfur application in kiwifruit, the SA content increased in leaves and the lignin content accumulated in stems [32]. Based on transcriptome data and qRT-PCR validation, we found that the expression level of *CsLAC37* increased under both *Pseudopestalotiopsis* species and *C. gloeosporioides* infections at different time points. Moreover, *CsLAC37* expression levels increased significantly in leaves at 9 h and 12 h under SA treatment. This evidence indicates that *CsLAC37* may be involved in resistance to fungal infections by triggering SA signaling pathways in tea plants.

To elucidate the function of *CsLAC37* in response to *C. gloeosporioides* infections, AsODN treatment was conducted in tea leaves. Although the average lesion sizes varied insignificantly between the AsODN and the controls under both normal and NBT/DAB staining conditions, the H_2_O_2_ content increased significantly, and POD activity decreased clearly. Reactive oxygen species (ROS) are critical for various abiotic and biotic stress sensing, the integration of multiple environmental signals, and the activation of stress-response pathways, so they are thereby helpful for the establishment of plant defense mechanisms [33]. H_2_O_2_ is a critical component of ROS, and its production increased when the host cell was damaged [34]. In plants, excessive endogenous ROS can damage host cells and peroxidases (PODs), which play a pivotal role in plant immunity by scavenging ROS accumulation [35,36]. Taken together, the results of AsODN treatment displayed that *CsLAC37* may play a positive regulatory role in resistance to *C. gloeosporioides* infections.

## 4. Materials and Methods

### 4.1. Plant Materials and Growth Conditions

Eight different tissues were collected from mature cloned ‘Shuchazao’ tea seedlings, including the apical bud, first leaf, second leaf, third leaf, young stem, young root, budding flower, and young fruit [37]. Mature tea plants were grown in the Tea Plant Cultivar and Germplasm Resource Garden (Hefei, China) of Anhui Agricultural University.

One-year-old cloned ‘Shuchazao’ tea seedlings were cultured in plastic pots (30 cm in diameter and 35 cm in height) and grown under a controlled culture environment (23 ± 3 °C, 65 ± 5% humidity, and 16/8 h day/light photoperiod) [8]. Seedlings with good health and uniform growth were sprayed with either 1 mM SA or 100 μM ABA until the leaves were completely wet, and leaves treated with distilled water served as the controls [38]. The second leaf of each branch was harvested at 3, 6, 9, 12, and 24 h after treatments.

*Pseudopestalotiopsis* species [18] and *C. gloeosporioides* [4] infections were caused in one-year-old cloned ‘Longjing 43’ tea seedlings, which were cultured in plastic pots with uniform growth. Moreover, one bud and two leaves were collected from one-year-old clonal ‘Longjing 43’ tea seedlings for gene suppression.

All treatments were conducted with three biological replicates, and samples were snap-frozen in liquid nitrogen and stored at −80 °C for further use.

### 4.2. Molecular Cloning and Bioinformatic Analysis

Based on the tea plant reference genome (http://tpia.teaplants.cn/) (accessed on 15 August 2022), we obtained the full-length genome sequence of *CsLAC37*. Then, gene-specific primers were designed to amplify the coding sequence of *CsLAC37* with cDNA templates of tea plant leaves. The amplified ORF was ligated into the pEASY-T1 plasmid and transformed into *E. coli* cells, then sequencing was performed. The primers used for cloning are listed in Appendix A. Amino acid sequences of AtLAC genes were collected from a website (https://www.arabidopsis.org/) (accessed on 3 September 2022). A phylogenetic tree was constructed based on the protein sequences using MEGA 11.0 with the neighbor-joining algorithm and 1000 replicates of bootstrap. The conserved domain of the CsLAC37 protein was analyzed by the website MEME (https://meme-suite.org/meme/tools/meme) (accessed on 3 September 2022). The molecular weight (MW) and isoelectric point (pI) were calculated by the ExPasy program (http://web.expasy.org/protparam/) (accessed on 4 September 2022). The full-length CDS and protein sequences are listed in Appendix A.

### 4.3. Subcellular Localization

The cloned ORF of *CsLAC37* was ligated into the pCAMBIA1305 vector to construct the fusion protein. The empty and constructed fusion expression vectors were transformed into *EHA105-*competent cells, then transiently introduced into tobacco (*Nicotiana benthamiana*) leaves by injection at 25 °C for 48 h in the darkness. GFP signals in the cells were examined by an Olympus FV1000 confocal microscope (Olympus, Tokyo, Japan). The relevant primers are listed in Appendix A.

### 4.4. RNA Extraction and qRT-PCR Analysis

Total RNA was extracted from the young leaves using the RC112-01 Kit (Vazyme, Nanjing, China). The quality of total RNA was detected using agarose gel electrophoresis, and its quantity was measured by a Nanodrop 2000 spectrophotometer (Thermo Fisher Scientific, Waltham, MA, USA). cDNA was transcribed from each RNA extract using the HiScript^®^ III RT SuperMix for qPCR (+gDNA wiper) kit (Vazyme, Nanjing, China). Detailed information of the reaction system was obtained and the process for qRT-PCR was performed as described previously [39]. *CsGAPDH* was used as the internal control, and the relative expression levels were analyzed using the 2^−△△Ct^ method [40]. All reactions were implemented in triplicate technical replicates, and three biological replicates were performed. The relevant primers are listed in Appendix A. The full-length nucleotide and protein sequences of the *CsLAC37* gene are listed in Appendix A.

### 4.5. Gene Suppression Using AsODNs

The candidate antisense oligonucleotide (AsODN) complementary sequences to the *CsLAC37* gene segments were selected by the Soligo online software 2.2 (https://sfold.wadsworth.org/cgi-bin/soligo.pl, accessed on 24 January 2024) [41]. Three AsODNs were synthesized and the probes for the *CsLAC37* are listed in Appendix A. Tender shoots with an apical bud and two leaves were excised from the tea plants and placed into 1.5 mL tubes containing 1 mL of 100 μmol AsODN-CsLAC37 [34]. The shoots in 1.5 mL tubes containing 1 mL of sense oligonucleotides (sODNs) and sterile deionized water were used as the controls. The second tender leaf at the same position of each shoot was harvested at 24 h and 48 h after treatments. Three independent biological replicates were conducted for each treatment, and at least three leaves were collected for each replicate.

### 4.6. Pathogen Inoculation and Treatment

The methods for pathogen inoculation and treatment followed our previous study [4] with minor modifications. Briefly, *C. gloeosporioides* were cultured on PDA solid medium and incubated for two weeks at 25 ± 2 °C in the darkness to promote sporulation. Conidia were collected and suspended in sterile distilled water, and the concentration was adjusted to 1 × 10^6^ conidia mL^−1^ for inoculation.

The second leaf from each shoot was surface-sterilized with 75% ethanol and sterile distilled water. The leaves were wounded with five pinholes (quincunx) on the leaf back surface using a sterile needle and inoculated with 50 μL of conidia suspension. An equal volume of sterile distilled water was used as a mock inoculation. Inoculated leaves were enclosed with plastic film and maintained in controlled conditions (25 ± 2 °C, 80% relative humidity, 16/8 h day and night). The *C. gloeosporioides*-inoculated and water-inoculated leaves were harvested at 3 d after treatments. All samples were frozen in liquid nitrogen immediately and stored at −80 °C. Leaf lesion size was measured using Image J 1.53t software. Three biological replicates were used for each treatment, and at least 10 leaves were inoculated for each replicate.

### 4.7. DAB and NBT Staining Assay

After gene suppression of *CsLAC37* for 24 h, tea leaves were inoculated with 50 μL conidia solution (10^6^ conidia mL^−1^). The second leaf of each shoot was collected to measure H_2_O_2_ and O_2_^−^ by performing diaminobenzidine (DAB) and nitroblue tetrazolium (NBT) staining, respectively. For H_2_O_2_ detection, the leaves were stained with 50 mM phosphate buffer (pH = 3.8) containing 1 mg/mL DAB, vacuumed for 1 h, and then incubated for 8 h at 37 °C in the darkness. For O_2_^−^ detection, the leaves were stained with 0.2% NBT in 50 mM phosphate buffer (pH = 7.5), vacuumed for 1 h, and incubated for 4 h at 37 °C in the darkness. Both the DAB- and NBT-stained leaves were immersed in 95% ethanol and boiled to eliminate chlorophyll.

### 4.8. Enzyme Activity Detection of POD and SOD

The gene-suppressed and infected leaves were also used for detection of peroxidase (POD) and superoxide dismutase (SOD) activities. The POD assay kit (A001-3-2) and SOD assay kit (A084-3-1) were used to detect POD and SOD activities, respectively, following the manufacturer’s instructions (Nanjing Jiancheng Bioengineering Institute, Nanjing, China). Briefly, a 1 g sample was ground in 9 mL sodium phosphate buffer (pH = 7.0) on ice, followed by centrifugation at 3500× *g* for 10 min at 4 °C and collection of the supernatant for further analysis.

## 5. Conclusions

The laccase family gene *CsLAC37,* involved in the response to *C. gloeosporioides* infections, was identified and cloned. *CsLAC37* had tissue-specific expression patterns, demonstrating that it had extremely high expression levels in stems and mature leaves. Subcellular localization showed that its protein was predominantly localized in the plasma membrane. *CsLAC37* was involved in responses to cold and salt stresses, and its expression levels increased significantly at some time points under SA and ABA treatments. Functional validation using AsODNs showed that *CsLAC37* may play a positive regulatory role in resistance to anthracnose disease.

## Figures and Tables

**Figure 1 plants-13-00884-f001:**
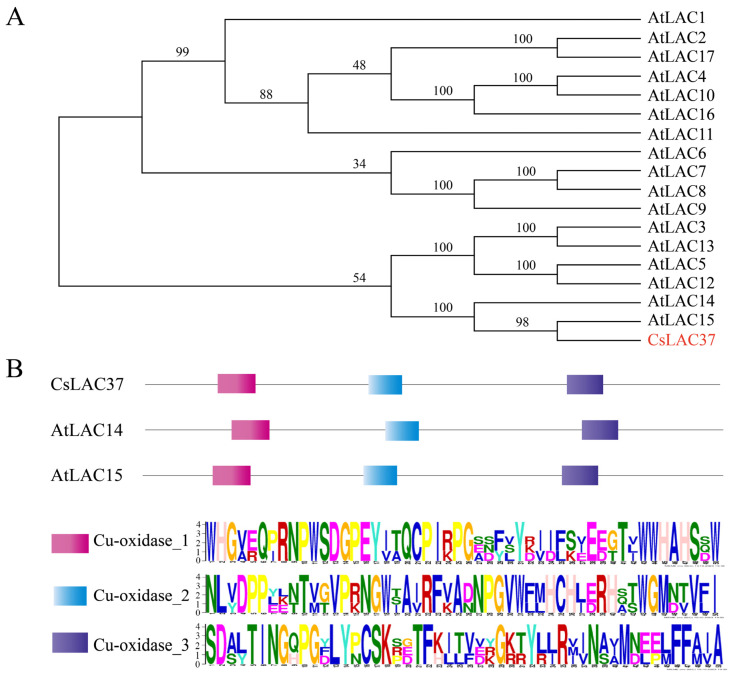
Identification and structural analysis of CsLAC37. (**A**) Phylogenetic tree based on amino acid sequences of *CsLAC37* and 17 *AtLACs* from *Arabidopsis thaliana*. The candidate gene *CsLAC37* from *Camellia sinensis* was marked in red. The phylogenetic tree was constructed by MEGA 11.0 with the neighbor-joining algorithm and a bootstrap of 1000. (**B**) Conserved domains of the CsLAC37, AtLAC14, and AtLAC15 proteins were visualized by the website MEME (https://meme-suite.org/meme/tools/meme) accessed on 30 October 2023. The red, blue, and purple rectangles represent Cu-oxidase_1, Cu-oxidase_2, and Cu-oxidase_3, respectively. The larger the letter, the higher the conservation of the site based on the website.

**Figure 2 plants-13-00884-f002:**
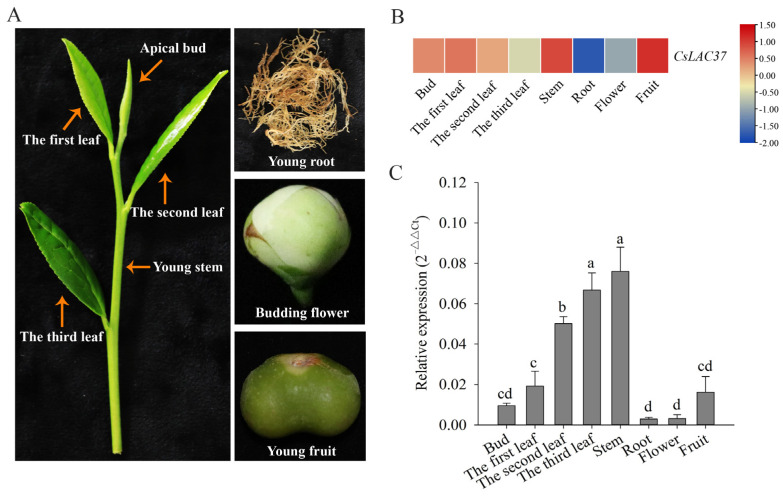
Tissue-specific expression pattern of *CsLAC37*. (**A**) Pictures of eight different tissues, including the bud, first leaf, second leaf, third leaf, young stem, young root, budding flower, and young fruit. (**B**) Heatmap of *CsLAC37* in eight different tissues based on transcriptome data. (**C**) Expression levels of *CsLAC37* in eight tissues based on qRT-PCR. A one-way analysis of variance with Duncan’s multiple-range test was performed. Different letters above the bars represent significant differences at *p* < 0.05.

**Figure 3 plants-13-00884-f003:**
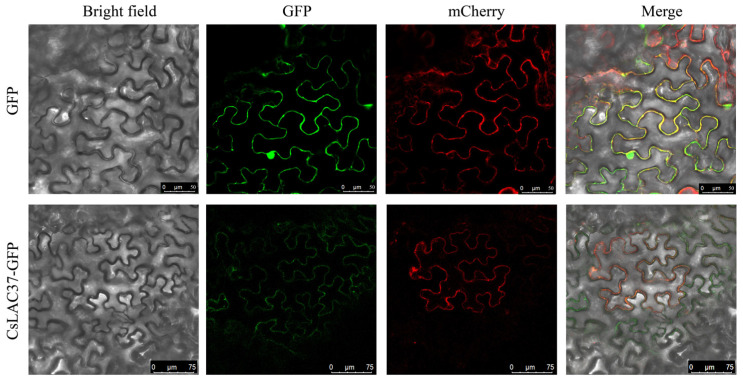
Subcellular localization of CsLAC37 protein. Scale bars represent 50 μm and 75 μm for pCAMBIA1305 (empty vector) and pCAMBIA1305-CsLAC37 expressed in the cells of tobacco leaves, respectively. Bright field, GFP, mCherry, and merge are shown for both empty vector and recombinant vector.

**Figure 4 plants-13-00884-f004:**
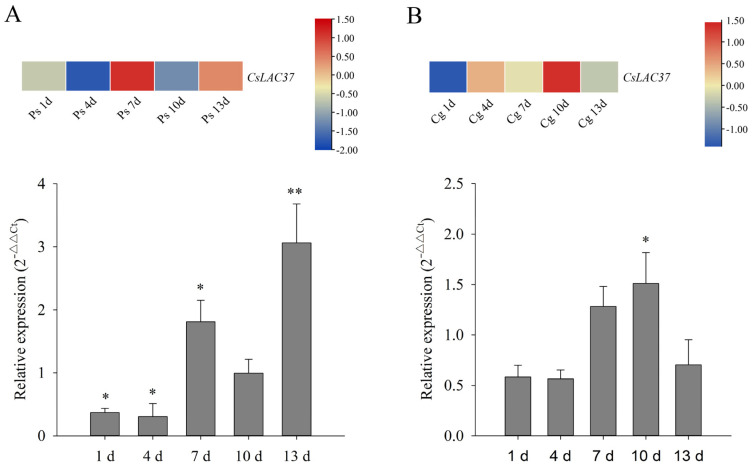
Expression analysis of *CsLAC37* in response to fungal pathogens. (**A**) Expression pattern of *CsLAC37* in response to *Pestalotiopsis-*like species infections based on transcriptome analysis and qRT-PCR. (**B**) Expression pattern of *CsLAC37* in response to *C. gloeosporioides* infections based on transcriptome analysis and qRT-PCR. The asterisks indicate the significance level (* *p* < 0.05 and ** *p* < 0.01) based on a Student’s *t*-test. Samples were harvested at 1, 4, 7, 10, and 13 d under both *Pestalotiopsis-*like species and *C. gloeosporioides* infections.

**Figure 5 plants-13-00884-f005:**
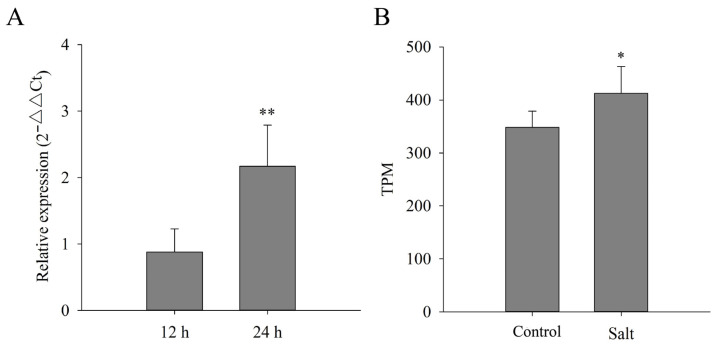
Expression analysis of *CsLAC37* under cold and salt treatments. (**A**) Relative expression level of *CsLAC37* under cold (4 °C) treatment by qRT-PCR. (**B**) Relative expression level (TPM value) of *CsLAC37* under salt treatment based on transcriptome data. Samples were collected at 12 and 24 h under cold treatment and 4 h under salt treatment. One or two asterisks indicate significant differences of *p* < 0.05 and *p* < 0.01, respectively, based on a Student’s *t*-test.

**Figure 6 plants-13-00884-f006:**
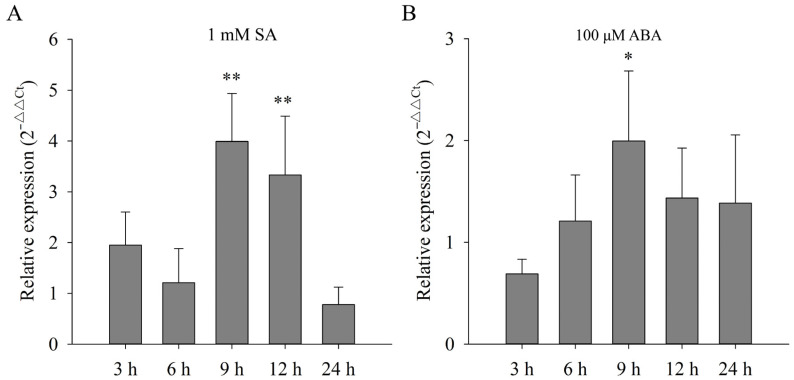
Expression analysis of *CsLAC37* under SA and ABA treatments. Relative expression level of *CsLAC37* (**A**) under SA and (**B**) ABA treatments. Samples were collected at 3, 6, 9, 12, and 24 h under both SA and ABA treatments. Based on a Student’s *t*-test, one or two asterisks represent significant differences of *p* < 0.05 and *p* < 0.01, respectively.

**Figure 7 plants-13-00884-f007:**
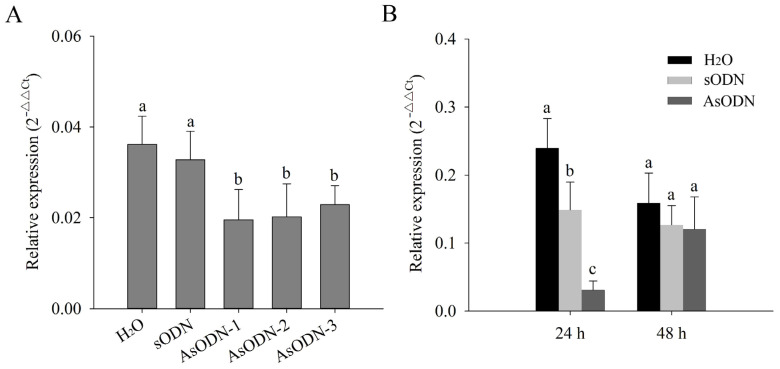
Selection of probes and optimal time points for antisense oligodeoxynucleotides (AsODNs). (**A**) Relative expression level of *CsLAC37* under AsODN by three different probes. (**B**) Relative expression level of *CsLAC37* at 24 h and 48 h under AsODN by probe AsODN-1. Both H_2_O and sODN were used as the controls. Different letters represent significant differences at *p* < 0.05 based on Duncan’s multiple-range test.

**Figure 8 plants-13-00884-f008:**
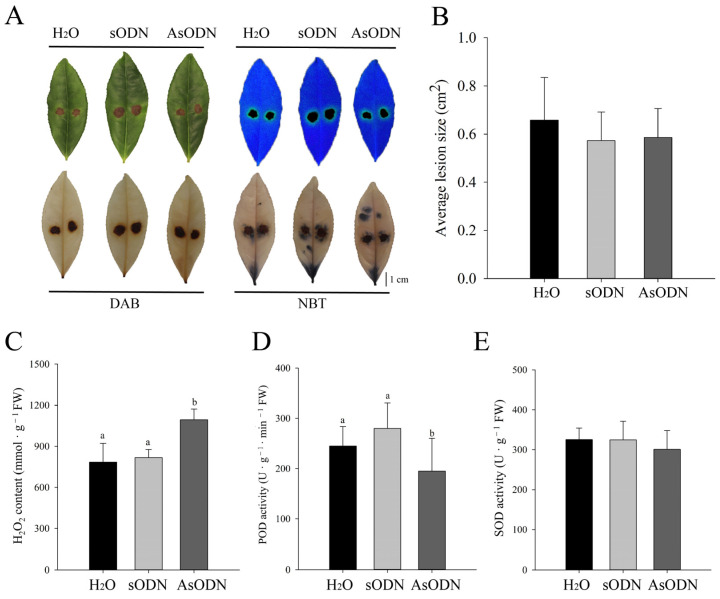
Functional validation of *CsLAC37* in response to *C. gloeosporioides* infection by AsODN treatment in tea shoots. (**A**) Leaves imaged under natural light (upper left corner), fluorescence (upper right corner), DAB staining (lower left corner), and NBT staining (lower right corner) after *C. gloeosporioides* infection by different treatments (H_2_O, sODN, and AsODN). (**B**) Statistics of the average lesion size in leaves after *C. gloeosporioides* infection by AsODN and the controls (H_2_O and sODN). (**C**) The content of H_2_O_2_ under AsODN and the controls. The (**D**) POD and (**E**) SOD activities under AsODN treatment and the controls. Different letters represent significant differences at *p* < 0.05 based on Duncan’s multiple-range test.

## Data Availability

All data supporting the findings are available within this paper and its Appendix A.

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
