# Peer review of "The Laccase Family Gene CsLAC37 Participates in Resistance to Colletotrichum gloeosporioides Infection in Tea Plants"

_plants, 2024, doi:10.3390/plants13060884_

Round 1

Reviewer 1 Report

Comments and Suggestions for Authors

In this study, the authors demonstrated CsLAC37, a laccase family gene may be involved in the resistance to anthracnose in tea plants. My comments/suggestions are as follows:

1.     Why did the authors use different tea plants in the SA- and ABA-associated experiments and the pathogen infection experiment?

2.     Line 18 and Line 97, “CDs” should be “CDS”. Line 25, “AsODN”, should give a full name when first mentioned. Line 283, sense oligonucleotides (sODN), should be “SODN”. Both “1 x 106 ml-1” and “106 conidia/ml” were used in this manuscript, which needs to be revised.

3.     Line 19, why “the highest homology”? there is no comparison, the same issue occurs in Line 75.

4.     Line 63-64, “We found that the expression level of CsLAC37 increased both under Pestalotiopsis-like species and C. gloeosporioides infection.”, here should be “…under both …”

5.     Sentences need to be rewritten because of grammar mistakes:

Line 67-68, “This study help us understanding the biological function of CsLAC37 and provide a candidate gene resource for tea plant breeding”.

Line 73-74, “The full-length ORF CsLAC37 was 1701 bp, and the protein with molecular weight of 68.1 kDa and isoelectric point (pI) value of 8.43”

Line 250, “The primers used for cloning were listed in Table S1”

Line 266-267, “The quality and quantity of total RNA were detected using agarose gel electrophoresis and a Nanodrop 2000 spectrophotometer”

6.     Bootstrap value needs to be added to each node in Figure 1B.

7.     In Figure 3, Scale Bars are unclear in some of the images.

8.     In Figure 4A, why is the data at 10 d anomalous compared to 7 d and 13 d?

9.     In Figure 5B, “CK” is not commonly used, it would be better to use “control” or “water” instead. In addition, data on this result come from other researchers’ experiments, but do not belong to this research, it would be better to put this result (and the corresponding description in Line 132-134) in the Discussion.

10.  Figure 6B, there were no differences except 9 h, as they have no difference, we cannot say “upregulated insignificantly” (Line 148).

11.  Line 158-160, “Subsequently, the AsODN-1 was selected to screen optimal time points for AsODN at 24 h and 48 h, indicating that the expression of CsLAC37 was suppressed significantly at 24 h compared to the controls (H2O and sODN) (Fig. 7B).” this illogical sentence needs to be rewritten.

12.  In Figure 8A, the leaves at the top and the bottom are not the same.

Comments on the Quality of English Language

grammar mistakes and illogical sentences.

Author Response

Below are our point-by-point responses to the reviewers’ comments. The revised sentences can be checked in the revised MS with changes.

Reviewer 1

In this study, the authors demonstrated CsLAC37, a laccase family gene may be involved in the resistance to anthracnose in tea plants. My comments/suggestions are as follows:

  1. Why did the authors use different tea plants in the SA- and ABA-associated experiments and the pathogen infection experiment?

Response: Thank you very much for your comments. Indeed, it would be better if the same tea cultivar was used for hormones-associated experiments and the pathogen infection experiment. In fact, the two experiments were performed at different time points, and it is difficult to obtain enough tea seedlings of the same tea cultivar at different time.

  1. Line 18 and Line 97, “CDs” should be “CDS”. Line 25, “AsODN”, should give a full name when first mentioned. Line 283, sense oligonucleotides (sODN), should be “SODN”. Both “1 x 106 ml-1” and “106 conidia/ml” were used in this manuscript, which needs to be revised.

Response: “CDs” was replaced by “CDS” (Line 19), the full name of AsODN was added (Line 28), both were revised as “106 conidia ml-1”. As for sense oligonucleotides, it should be sODN (Wang, et al. 2021, Plant J,106,862-875; Zhao et al. 2019, New Phytologist, 226, 362-372; Jing et al. 2023, Plant Cell and Environment, 1-16; etc.)

  1. Line 19, why “the highest homology”? there is no comparison, the same issue occurs in Line 75.

Response: Thank you for your comments. They were revised as “…had the closest relationship with the Arabidopsis AtLAC15 protein compared to other AtLACs” (Line 21) and “… had the highest similarity with AtLAC14 and AtLAC15 compared to the other AtLACs (Line 95), respectively.

  1. Line 63-64, “We found that the expression level of CsLAC37 increased both under Pestalotiopsis-like species and C. gloeosporioides infection.”, here should be “…under both …”

Response: It was revised accordingly (Line 81).

  1. Sentences need to be rewritten because of grammar mistakes:

Line 67-68, “This study help us understanding the biological function of CsLAC37 and provide a candidate gene resource for tea plant breeding”.

Line 73-74, “The full-length ORF CsLAC37 was 1701 bp, and the protein with molecular weight of 68.1 kDa and isoelectric point (pI) value of 8.43”

Line 250, “The primers used for cloning were listed in Table S1”

Line 266-267, “The quality and quantity of total RNA were detected using agarose gel electrophoresis and a Nanodrop 2000 spectrophotometer”

Response: Thank you very much for your suggestions. Please check the revised sentences below.

Line 84-86. “This study helps us understand the biological function of CsLAC37 and provides a candidate gene resource for tea plant breeding”.

Line 92-94. “….demonstrating that the full-length ORF CsLAC37 was 1701 bp, and the protein had a molecular weight of 68.1 kDa and isoelectric point (pI) value of 8.43”.

Line 314-315. The primers used for cloning are listed in Table S1

Line 331-332. The quality of total RNA was detected using agarose gel electrophoresis and its quantity was measured by a Nanodrop 2000 spectrophotometer.

  1. Bootstrap value needs to be added to each node in Figure 1B.

Response: Bootstrap value had been added in the revised Figure 1A.

  1. In Figure 3, Scale Bars are unclear in some of the images.

Response: Scale bars had been revised.

  1. In Figure 4A, why is the data at 10 d anomalous compared to 7 d and 13 d?

Response: Thank you for your question. The data was confirmed based on transcriptome data and qRT-PCR results.

  1. In Figure 5B, “CK” is not commonly used, it would be better to use “control” or “water” instead. In addition, data on this result come from other researchers’ experiments, but do not belong to this research, it would be better to put this result (and the corresponding description in Line 132-134) in the Discussion.

Response: Thank you for your suggestions. “CK” was replaced by “Control”. Actually, it is very important to know that the CsLAC37 was involved in the response to salt stress. This result was retained and the reference was cited clearly.  

  1. Figure 6B, there were no differences except 9 h, as they have no difference, we cannot say “upregulated insignificantly” (Line 148).

Response: Thank you. We revised it as “…while no significant variations were found …” (Line 186).

  1. Line 158-160, “Subsequently, the AsODN-1 was selected to screen optimal time points for AsODN at 24 h and 48 h, indicating that the expression of CsLAC37 was suppressed significantly at 24 h compared to the controls (H2O and sODN) (Fig. 7B).” this illogical sentence needs to be rewritten.

Response: Subsequently, the AsODN-1 was selected to screen optimal time points for AsODN after 24 h and 48 h. The results showed that the expression level of CsLAC37 was suppressed significantly after 24 h compared to the controls (H2O and sODN), while no significant difference was observed after 48 h.

  1. In Figure 8A, the leaves at the top and the bottom are not the same.

Response: Sure, different leaves were used for different staining methods (DAB and NBT).

Reviewer 2 Report

Comments and Suggestions for Authors

Li et al submitted a manuscript titled "The laccase family gene CsLAC37 participates in resistance to Colletotrichum gloeosporioides infection in tea plants" for publication in Plants.

Though the significance of research in this area is high, and the work fits within the scope of this journal, there are few very important considerations the authors have to address before the ms could be reviewed again and decided for its acceptance.

Almost all figure descriptions are sub-par. I urge authors to read previously accepted publications to understand the framework of a figure description. Each figure description should have a theme title for the part of results it presents. Describe methods used in brief to generate these results. Describe every tiny component of each figure.

Fig. 1A Gel picture is not required. It looks very rudimentary to show length on a gel.

1B describe tools used to generate this tree and describe what a reader should infer from this tree, and describe what red font represents.

1C Describe what are pink, cyan and purple blocks represent, and what should a reader infer from lower panel. Describe weblogo.

1D why is tertiaty structure shown? How is it helping understand the results or meeting objectives? How is it generated, and what colors represent? If it is kept just as another analysis of an obtained sequence, delete it.

Fig 2C. Mention the tests used to determine significance; here and elsewhere applicable.

Follow these guidelines in all other figure/table descriptions. Authors should remember that a reader should understand what is presented in any one set of figures (standalone) without reading results section or the whole paper.

Comments on the Quality of English Language

English language is fine. Use established nomenclature or abbreviations.

For eg. Line 18 - CDs is wrong. It should be CDS. Modify here and elsewhere.

Author Response

Reviewer 2

Li et al submitted a manuscript titled "The laccase family gene CsLAC37 participates in resistance to Colletotrichum gloeosporioides infection in tea plants" for publication in Plants.

Though the significance of research in this area is high, and the work fits within the scope of this journal, there are few very important considerations the authors have to address before the ms could be reviewed again and decided for its acceptance.

Almost all figure descriptions are sub-par. I urge authors to read previously accepted publications to understand the framework of a figure description. Each figure description should have a theme title for the part of results it presents. Describe methods used in brief to generate these results. Describe every tiny component of each figure.

General reply: Thank you very much for your constructive comments. We have made careful revisions in the revised MS based on your comments. English writing was also polished by an English editing service. Below are our point-by-point responses to your comments and the revised sentences can be checked in the revised MS with changes.

  1. 1A Gel picture is not required. It looks very rudimentary to show length on a gel.

Response: Fig. 1A and the corresponding descriptions have been deleted.

  1. 1B describe tools used to generate this tree and describe what a reader should infer from this tree, and describe what red font represents.

Response: The bootstrap values were added in Fig 1B and the detailed description of figure legends was supplemented. Detailed information please check the revised MS (line 118-124).

  1. 1C Describe what are pink, cyan and purple blocks represent, and what should a reader infer from lower panel. Describe weblogo.

Response: Thank you for your suggestions. A description was added, such as “The red, blue, and purple rectangles represent Cu-oxidase_1, Cu-oxidase_2, and Cu-oxidase_3, respectively”. The larger the letter, the higher conservation of the site based on the weblogo.

  1. 1D why is tertiaty structure shown? How is it helping understand the results or meeting objectives? How is it generated, and what colors represent? If it is kept just as another analysis of an obtained sequence, delete it.

Response: Fig. 1D and its corresponding description were deleted. Figure 1 contained phylogenetic tree (1A) and conserved motifs (1B).

  1. Fig 2C. Mention the tests used to determine significance; here and elsewhere applicable.

Follow these guidelines in all other figure/table descriptions. Authors should remember that a reader should understand what is presented in any one set of figures (standalone) without reading results section or the whole paper.

Response: Thank you for your suggestions. Figure 2 legend was supplemented. Such as “Tissue-specific expression analysis of CsLAC37. (A) Pictures of eight different tissues, including bud, the first leaf, the second leaf, the third leaf, young stem, young root, budding flower, and young fruit. (B) Heatmap of CsLAC37 in eight different tissues based on transcriptome data. (C) Expression levels of CsLAC37 in eight tissues based on qRT-PCR. Different letters above the bars represent significant differences at P < 0.05.”

  1. For eg. Line 18 - CDs is wrong. It should be CDS. Modify here and elsewhere.

Response: “CDs” was replaced by “CDS” in the whole manuscript.

Round 2

Reviewer 1 Report

Comments and Suggestions for Authors

Thanks for addressing my comments, I have a few more questions as follows:

1.     The present simple tense and past simple tense are confusing in this manuscript, and also singular and plural of some nouns.

2.     The authors did not answer my 8th question by saying “The data was confirmed based on transcriptome data and qRT-PCR results.

3.     The authors did not answer my 12th question by saying “different leaves were used for different staining methods (DAB and NBT). Of course, I know you certainly did not use one leaf for both DAB and NBT. I mean it will be better to show the same leaf before and after the DAB/NBT.

4.     Line 138-140, “The expression levels of CsLAC37 increased significantly after 7 d and 13 d under grey blight infection based on transcriptome data, while it decreased dramatically after 4 d”, why did the authors change “at” to “after”? I think they checked at 4 d, 7d, and 13 d. While it decreased after 4 d, it is paradoxical when they say it increased significantly after 7 d and 13 d. If the authors mean to say after the infection, I suggest using “dpi”.

Comments on the Quality of English Language

The present simple tense and past simple tense are confusing in this manuscript, and also singular and plural of some nouns.

Author Response

Reviewer 1

Thanks for addressing my comments, I have a few more questions as follows:

General response: Thank you very much for your critical comments. We carefully checked and revised the whole MS according to your suggestions. The revised sentences or words were marked in red. Below are our point-by-point responses to your comments.

  1. The present simple tense and past simple tense are confusing in this manuscript, and also singular and plural of some nouns.

Response: We checked the MS carefully and revised the wrong words or sentences that were marked in red. Detailed information please check the revised MS.  

  1. The authors did not answer my 8th question by saying “The data was confirmed based on transcriptome data and qRT-PCR results.”

Response: Thank you. In figure 4A, both transcriptome data and RT-PCR results showed that the CsLAC37 expression level was varied insignificantly at 10 d after fungal infection compared to the control. We speculate that CsLAC37 was upregulated at the most of time points after 4 dpi, but did not exclude it varied insignificantly at some time points.

  1. The authors did not answer my 12th question by saying “different leaves were used for different staining methods (DAB and NBT). Of course, I know you certainly did not use one leaf for both DAB and NBT. I mean it will be better to show the same leaf before and after the DAB/NBT.

Response: Thank you for your advice. Indeed, it would be better if the same leaf was used for staining after took photos. However, we used different tea leaves for taking photos and DAB/NBT staining. The same experiment will be improved following your advice for the next time.

  1. Line 138-140, “The expression levels of CsLAC37 increased significantly after 7 d and 13 d under grey blight infection based on transcriptome data, while it decreased dramatically after 4 d”, why did the authors change “at” to “after”? I think they checked at 4 d, 7d, and 13 d. While it decreased after 4 d, it is paradoxical when they say it increased significantly after 7 d and 13 d. If the authors mean to say after the infection, I suggest using “dpi”.

Response: Thank you for your comments. Actually, the English editing service suggest we change “at” to “after”, thereby we changed it. Based on your advice, we replaced “after” by “at” in the whole MS. Please check the revised MS.